# Parental Knowledge, Attitude, and Practices on Antibiotic Use for Childhood Upper Respiratory Tract Infections during COVID-19 Pandemic in Greece

**DOI:** 10.3390/antibiotics10070802

**Published:** 2021-07-01

**Authors:** Maria-Eirini Oikonomou, Despoina Gkentzi, Ageliki Karatza, Sotirios Fouzas, Aggeliki Vervenioti, Gabriel Dimitriou

**Affiliations:** Department of Paediatrics, Medical School, University of Patras, Rion 26504, Greece; mareiroik@yahoo.gr (M.-E.O.); akaratza@hotmail.com (A.K.); sfouzas@upatras.gr (S.F.); aggelikivervenioti@gmail.com (A.V.); gdim@upatras.gr (G.D.)

**Keywords:** upper respiratory tract infections, antibiotics, antimicrobial resistance, parental knowledge, attitudes, practices, child, Greece, COVID-19

## Abstract

This cross-sectional study aims to assess parents’ knowledge, attitude, and practices on antibiotic use for children with URTIs symptoms in Greece in the era of the COVID-19 pandemic. We distributed a questionnaire to a random sample of parents who visited primary health care centers in Patras, Greece. Out of 412 participants, 86% believed that most infections with common cold or flu symptoms were caused by viruses, although 26.9% believed that antibiotics may prevent complications. Earache was the most common symptom for which antibiotics were needed. Most of them (69%) declare being considerably anxious about their children’s health during the COVID-19 pandemic. The majority (85%) knew that COVID-19 was of viral origin, yet half of them declared uncertain whether antibiotics were needed. All demographic characteristics, except for gender, were found to have a significant effect on parents’ knowledge, attitude, and practices on antibiotic use for URTIs and COVID-19. Factor analysis revealed six groups of parents with common characteristics associated with misuse of antibiotics. Our findings highlight the need to decrease misconceptions regarding antibiotic use by providing relevant education for parents targeting particular characteristics, especially during the COVID-19 pandemic. Continuous education of healthcare providers in the field is also of paramount importance.

## 1. Introduction

Upper respiratory tract infections (URTIs) in children account for more than 10% of the total outpatient and emergency department visits and are considered the main cause of children’s absenteeism from school as well as parents’ from the workplace [1,2]. The discovery of antibiotics is a very important achievement of the 20th century, and their proper and judicious use reduces mortality and morbidity [3]. However, antibiotics are often used inappropriately, especially for the treatment of URTIs in children, even though there is sufficient evidence to support the viral origin of these conditions and the fact that this practice does not change the duration and severity of symptoms [4,5]. Many issues have been raised because of overuse or misuse of antibiotics, such as the high cost of health services, rise of side effects such as diarrhea, and increased antibacterial resistance [6,7,8]. There is a strong association between antibiotic overuse and the development of resistance, and countries with high rates of antibiotic consumption have reported a high incidence of resistant pathogens [9,10]. As a result, over the past decade, the World Health Organization (WHO) recognized the emergence of antibiotic resistance as one of the 10 top global public health threats facing humanity [11]. Based on numerous reports, factors leading to antimicrobial overuse in children are complex, involving easy access to antibiotics for self-medication [12], overprescribing by physicians due to diagnostic uncertainty [13], or parents’ wish, which is mainly driven by lack of knowledge and misperceptions [14,15,16,17]. A recent review demonstrated that one-third of the population of low and middle-income countries have significant knowledge gaps in the field [18]. The quality of communication between parents and pediatricians plays an important role in the decision of prescribing [19]. Associations between demographic characteristics and parental knowledge, attitude, and practices towards antibiotic use in URTIs have also been described [13,20,21,22,23,24].

Coronavirus infectious disease 2019 (COVID-19), declared as a pandemic on the 11th of March of 2020, may cause URTI symptoms and, in the majority of children, warrants only symptomatic treatment [25,26,27]. No studies on parents’ knowledge, attitude, and practices on antibiotic use have been thus far conducted during the pandemic. Despite numerous national educational activities in Greece on judicious antibiotic use, antibiotic consumption in the community has been found to be the highest amongst 31 European countries [28,29,30]. A nationwide study conducted in 2010 demonstrated that Greek parents understand the benign course of URTIs, rarely give antibiotics without medical advice, and contribute less than expected to antibiotic misuse [31]. The present study aims to explore parental knowledge, attitude, and practices (KAP) regarding antibiotic use for URTIs and COVID-19 infection in Greece, during the COVID-19 pandemic, with the view of designing educational strategies focusing on those parents who are prone to antibiotic misuse.

## 2. Results

### 2.1. Parents’ Demographic Characteristics

Out of the 499 questionnaires distributed, 412 were returned (response rate 82.5%). Table 1 shows the demographics of our studied population. Of note, the majority of our study participants were women (77.2%), aged between 31–50 years old (88.1%), of Greek nationality (85.7%), and with good self-perceived access to the health care system (65.8%).

### 2.2. Parental Knowledge on Antibiotic Use in Children with URTIs

The vast majority of parents (81.6%) knew that infections with signs of cold or flu symptoms were caused by viruses. Most of them acknowledged that antibiotics were not necessary, and (69%) answered that symptomatic treatment was only needed for viral infections. Of note, 48.5% of parents thought that antibiotics were needed for bacterial infections while 50.7% for viral ones. Earache was the most common symptom for which parents thought that antibiotics were needed (25.2%), followed by a runny nose (9.5%). More than half of the parents (52.2%) declared that antibiotics were always needed for their child’s tonsillitis and 47.1% for otitis. The majority of parents (73.6%) identified successfully antibiotics between other drugs. Half of them did not believe that antibiotic use may protect them from complications of the common cold or the flu. Most of the parents (80.8%) knew that imprudent use of antibiotics reduces their efficacy and may cause antibiotic resistance, and two-thirds of them were familiar with the fact that antibiotics may have some side effects. Most parents (73.3%) declared to be adequately informed about judicious use of antibiotics, and 91.7% of them declared that an educational program about the proper use of antibiotics would be useful.

### 2.3. Parental Attitude and Practices towards Antibiotic Use in Children with URTIs

Nearly half of the parents (48.8%) declared that they did not know whether their child needed an antibiotic before visiting the pediatrician. Most of the responders (69.4%) would not feel more satisfied if their pediatrician were to prescribe antibiotics for their child’s cold symptoms. Moreover, 93% of the participants always comply with their pediatrician’s recommendations. When asked if they would visit another pediatrician for a second opinion in case their own would not prescribe antibiotics, 82% disagreed. In addition, the majority (86.6%) would not change pediatrician because they would not prescribe antibiotics as frequently as they desired. Most of them never used leftover antibiotics, while 13.6% sometimes did. A total of 84% of the participants never insisted that their pediatrician prescribe antibiotics, and most of them (90.3%) answered that their pediatrician did not prescribe antibiotics because they insisted, while 7.8% declared that this happens frequently. A total of 85% did not purchase antibiotics for their child’s cold or flu symptoms over the counter, while 15% of them adopted this practice sometimes, often, or always. Moreover, 61.4% of the parents answered that their pediatrician never recommended antibiotics over the phone, while 32.5% answered that this had happened sometimes. If their child was unwell, parents always (50.5%) or most of the time (27.6%) consulted their pediatrician, followed by previous experience where they referred to sometimes (27.9%) or often (19.7%). Moreover, 23.1% sometimes asked their relatives, while 14.6% sought advice from the internet.

### 2.4. Parental Knowledge Attitude, and Practices towards Antibiotic Use in Children with COVID-19 Infection

Most parents (85%) declared that COVID-19 infection was of viral origin, and 72.3% answered that it may cause flu and cold symptoms in children. Most parents (68.4%) declared being anxious for their child’s health during the COVID-19 pandemic. Approximately half of the parents (49.8%) declared unaware whether antibiotics were needed for COVID-19, and 47.8% did not know whether their child would be sick for a longer time if they would not receive antibiotics for COVID-19. Half of the parents (51.5%) declared more satisfied if their pediatrician would not recommend antibiotics for their child’s cold or flu symptoms. The vast majority of parents (93.2%) answered that their pediatrician did not recommend antibiotics over the phone for their child’s cold or flu symptoms during the pandemic.

### 2.5. Effect of Demographic Characteristics and Knowledge, Practice, and Attitude on Antibiotic Use

The correlations between sociodemographic characteristics and KAP on antibiotic use during the COVID-19 pandemic are shown in Table 2. Of note, KAP on antibiotic use during COVID-19 was not associated with parental age and gender, chronic disease in the child, and number of children in the family. Various other associations have been demonstrated between KAP and other sociodemographics. The relevant correlations between KAP and URTIs in general (not only for COVID-19) are available as Appendix A.

Table 3 shows the results of factor analysis. We found six groups of parents who misused antibiotics, which in total accounts for 76.6% of the variation of the answers given on KAP in our study (meaning that 76.6% of our study participants belong to at least one of these groups).

## 3. Discussion

This is the first study on knowledge, attitude, and practices from parents in Greece regarding antibiotic use in young children with URTIs in the era of the COVID-19 pandemic. The response rate was 82.4%, which is rather satisfactory, especially taking into account the COVID-19 strict restrictions during visits in primary health care.

According to the results of this study, parents have a satisfactory level of knowledge on antibiotic use for URTIs, however some incorrect perceptions have been revealed. The majority of parents (81.6%) correctly answered that most URTIs are viral in origin, in contrast to parents from a similar study from Northern Ethiopia who scored lower percentages of correct answers (31.4%) [15]. In the latter study, 30% of the responders believed incorrectly that antibiotics were needed for viral infections, and 20.6% declared to be uncertain. Of note, this percentage is even smaller than other studies in China and Pakistan (79% and 83%, respectively) [32,33]. Almost 70% of parents were aware that common cold and flu need only supportive care, although 26.9% falsely believed that antibiotic use may protect from complications while 26.9% declared uncertain, which are similar percentages to those previously reported [32,33,34,35,36,37,38,39]. A total of 15.6% answered that their children’s disease will get complicated in the case of not using antibiotics in contrast to similar surveys in China [32], Cyprus [14], and the Republic of Macedonia [18], in which the percentages were higher, 50%, 48%, and 44%, respectively. In our study, most responders were familiar with antibiotics and able to distinguish them among other drugs similar to Palestinian parents [38]. In particular, 61% were worried about antibiotics’ side effects, and the majority knew that imprudent use drives antimicrobial resistance, as previously depicted in similar studies worldwide [8,14,16,23].

The results of this study regarding parents’ attitudes and practices towards antibiotic use for children’s URTIs are of great interest. Parents do not seek antibiotics as 83% of them do not insist on this practice, and 69% are not more satisfied if this happens. Less than 10% believed that their pediatrician prescribes antibiotics due to their insistence, which is less than Palestinian parents (28%) but more than the percentage of parents from Cyprus [8,14]. Of note, only 8% of Australian parents asked their pediatrician for antibiotics (8%), whereas 82% of Chinese parents mentioned that they would not get displeased if their pediatrician rejected their demand for antibiotics [22,32]. In general, parents expect physical examination and adequate consultation for the nature of the disease and the need for antibiotics [37,38,39,40]. These findings emphasize the vital role of communication between parents and pediatricians.

The findings of the survey also demonstrate that Greek parents do not treat their children without medical advice. Only 12.6% of the participants reported that they have sometimes purchased antibiotics without medical prescription by the drug store. This finding is lower than in other countries such as Peru (23.5%) and Lebanon (22.5%) [34,35,36,37,38]. Of note, 16.3% have sometimes used leftover antibiotics in contrast with Saudi Arabia and the Republic of Macedonia, where higher percentages were recorded, 66%, 60.8%, respectively [16,18]. Moreover, it appears that Greek parents trust their pediatricians as the vast majority would neither visit another pediatrician for a second opinion (81.3%) if their pediatrician would not prescribe antibiotics for their child’s cold or flu, nor change their pediatrician in case they did not prescribe antibiotics as often as they believed that they should (86.4%). In addition, 93% always comply with their pediatrician recommendations in regards to their child’s cold or flu symptoms. These findings are similar to a similar study in Palestine but differ from a study in Jordan where 39.3% of the parents answered that they would indeed change pediatrician if they prescribed antibiotics less frequently than they desired [23,39]. The pediatrician remains the main source of information for the ill child, followed by previous experience, pharmacist, relatives, and the internet. Finally, more than one-third of the parents obtained antibiotics for their child following a telephone consultation, and the same percentages have been reported in Jordan and Cyprus [14,23].

It is of interest to note that we did not find many differences in comparison with a similar survey conducted in our country a decade ago [31]. Earache was the most common symptom for which parents anticipated receiving antibiotics. Whereas in both studies, most parents declared to be aware of antimicrobial resistance due to antibiotic misuse (88% in the previous and 81% in our study), a similar percentage of parents (24.7% in the previous and 24% in our study) would administer antibiotics in case their child got flu or cold symptoms in order to recover sooner. Moreover, similar percentages of parents (42% in the previous and 38% in the current study) got a medical recommendation for antibiotics over the phone. As a result, the development and implementation of parental education measures to improve antibiotic use are crucial and still warranted at a national level.

This is the first study on parental knowledge, attitude, and practices towards antibiotic use on COVID- 19 infection in children in our country. The majority of parents (68.4%) declared to be extremely anxious as far as their children’s health is concerned due to the pandemic. Most parents knew that COVID-19 causes flu or cold symptoms and is of viral origin. However, half of them declared to be uncertain as to whether antibiotics were needed for its treatment, and a third of them were not aware if their child’s illness would last for longer if no antibiotics are used. These results were predictable as COVID-19 was a new infection, and parents were not adequately informed, especially in the beginning of the pandemic. Therefore, pediatricians should invest more time in order to inform parents about COVID-19. As far as over the telephone diagnosis and recommendation of antibiotics are concerned, the vast majority of parents (93%) replied that this never happened during the pandemic, whereas 63% replied that this was never practiced by their pediatrician before. Parents expect physical examination and recommendation of the appropriate treatment. Given the circumstances during the pandemic, this might not always be feasible. Healthcare providers should therefore have a low threshold of antibiotic prescribing and continue to adhere to the antimicrobial stewardship principles during the pandemic to combat further development of resistant pathogens. 

Factor analysis revealed six groups of parents with common characteristics that were associated with antibiotic misuse. Group 1 that accounts for 24% of the variation includes parents that do not have an adequate background on judicious antibiotic use. Even worryingly, group 2 (10.2% of the variation) includes parents that are confident in antibiotic use yet misuse them. For these two groups, more time needs to be invested by prescribers in educating parents in the field. On the other hand, groups 3, 4, 5, and 6 (31.5% of the variation) are parents that do not use antibiotics prudently but do listen to their pediatricians. The latter highlights the need for interventions on antibiotic prescribers, including continuing education. Educational activities for prescribers should start early on in their career, as research thus far has highlighted gaps in medical training in the field [41,42]. Novel and effective training methods on all aspects of antimicrobial stewardship and antimicrobial resistance should be incorporated in the Medical Curricula worldwide. In addition to education, better prescribing requires expert personnel, practice guideline drafting, and implementation aids, as well as setting clear goals and quantitative targets for quality in prescribing [43]. All the above have been used with apparent success in some Northern European countries and should guide relevant interventions in Greece [43]. 

It is of interest to highlight that our study population included a small percentage of parents with low cultural and economic levels. In particular, more than half of our study participants were urban with a university education. As the problem of antimicrobial misuse is a social issue, it demands a relevant approach from that perspective [44]. Hence, a good level of knowledge on antibiotics use could be partially related to the high socioeconomic status of our participants. The involvement of social scientists in the field is crucial to address the multiple dimensions of the problem, such as socio-cultural, economic, and political. In addition to the above, given the dynamic and constantly changing nature of human behavior, social media could also play an important role in public health interventions to combat antimicrobial resistance, and their role is yet to be further established [44,45]. 

We acknowledge that our study as a pragmatic one has strengths and limitations. One of its strengths is that data were collected through self-administered questionnaires. This methodology was preferred more than face-to-face or telephone interviews in order to avoid the chance of the interviewer influencing the parent’s response as well as the likelihood that interviewers respond in accordance with what is believed to be the expected answer. With regards to study limitations, firstly, the distribution of the questionnaires took place in the summer months of 2020 (during precautionary measures to prevent the spread of COVID-19) with a low number of primary care visits. Secondly, it was difficult for parents with low socioeconomic status or belonging to special population groups (e.g., Roma) to take part in the study because they might have been incapable of reading or understanding the questionnaire. Moreover, the study enrolled parents who only sought care at health care centers, which limits the generalizability of the results. Finally, data were collected from a large city in Greece and thus mainly represent the urban but not necessarily the rural population of the area.

## 4. Materials and Methods

We performed a cross-sectional study including parents of children aged 0–16 years and attending the primary pediatric healthcare services in the city of Patras during a period of 4 months (May–August 2020). Patras is the third-largest city in Greece after Athens and Thessaloniki. All parents/guardians of children attending the primary care clinics were informed in detail with a written information sheet about the study aims, as well as data confidentiality. Parents who agreed to take part in the study were asked to complete an anonymous self-administered questionnaire regarding their KAP on the use of antibiotics for childhood URTIs and COVID-19. The questionnaire was written in Greek and consisted of 2 parts. Part A included 13 questions on demographics (age, gender, nationality, level of education, occupation, health insurance status, single-family status, child’s chronic disease, access to the healthcare system). Part B included 23 questions on the parent’s knowledge, attitudes, and practices on antibiotic use for URTIs and COVID-19. Answers were given on a 5-point Likert scale (“strongly agree”, “agree”, “uncertain”, “disagree”, “strongly disagree”, or “never”, “sometimes”, “often”, “most of the time”, “always”). Statistics were performed using the SPSS v.21 software (IBM Corp, Armonk, NY, USA). The level of significance was set to 0.05 for all analyses. Descriptive statistics were used to summarize data on demographic characteristics, knowledge, attitude, and practices towards antibiotic use. Chi-square test was used to test for significant association between demographic characteristics and parents’ knowledge, attitude, and practices. Unsure responses were scored as incorrect. Factor analysis was used in order to identify groups of parents with common characteristics that were at high risk of inappropriate antibiotic use.

## 5. Conclusions

This study is an important step towards a better understanding of the knowledge, attitude, and practices regarding antibiotic use in URTIs in children during the COVID-19 pandemic. It revealed that there was a diversity of parental awareness about antibiotics and antimicrobial resistance on the basis of social-demographic factors. Of note, the pandemic does not seem to have influenced parental views on antibiotic use for their offspring. Educational campaigns targeting high risk groups to decrease misconceptions about antibiotic use and to increase awareness regarding the risks of inappropriate use are warranted. Targeted education on the particularities of COVID-19 is also vital for parents. In addition, continuous training of pediatricians about effective communication with parents and prudent prescribing of antibiotics is essential. Ideally, this study should be conducted with a large country-representative sample and during the consequent phases of the pandemic in order to assess for possible changes. It would also be useful to assess attitudes pre and post-training, specifically designed for that purpose, for both parents and pediatricians. Furthermore, studies assessing country-specific determinants of antibiotic misuse, such as country wealth and health care system particularities, would also be useful for the implementation of multilevel interventional programs aimed at limiting the spread of antibiotic resistance. Despite multiple efforts to achieve that in Greece during the last decades, there is still room for further development. Public health interventions at a national level should be constant and sustainable following the successful examples of other European countries. Finally, antimicrobial stewardship activities and interventions should not be neglected during the COVID-19 pandemic.

## Figures and Tables

**Table 1 antibiotics-10-00802-t001:** Demographic characteristics of the study participants.

Demographic Characteristics	Total Frequency (*n* = 412)	Percent (%)
**Gender**		
Female	318	77.2
Male	91	22.1
Ν/A *	3	0.7
**Parental age (years)**		
18–30	26	6.3
31–50	363	88.1
51–60	20	4.9
N/A	3	0.7
**Ethnicity**		
Greek	353	85.7
Non—Greek	59	14.3
**Education**		
Primary school	6	1.5
Secondary school	20	4.9
High school	76	18.4
College	42	10.2
University	170	41.3
Postgraduate-doctoral degree	59	14.3
N/A	39	9.5
**Occupation**		
Unemployed	87	21.1
Student	2	0.5
Part-time	34	8.3
Full-time	287	69.7
Retired	2	0.5
**Medical Insurance**		
Public	358	86.9
Private	13	3.2
Both	19	4.6
None	22	5.3
**Family income**		
Low	75	18.2
Middle	311	76.9
High	20	4.9
**Age of child**		
0–5 years	121	29.4
6–11 years	166	40.3
12–16 years	122	29.6
N/A	3	0.7
**Single-parent family**		
Yes	36	8.7
No	376	91.3
**Chronic disease in child**		
Yes	19	4.6
No	393	95.3
**Access to the healthcare system**		
Low	40	9.7
Moderate	101	24.5
Good	271	65.8
**Pediatrician**		
Public	259	62.9
Private	153	37.1

* N/A: not answered.

**Table 2 antibiotics-10-00802-t002:** Association of demographic characteristics with knowledge/attitude/practice statements on COVID-19.

Statement/Question	Correct	Incorrect	*p*-Value (Chi-Square Test)							
			Nationality	Education	Occupation	Insurance	Income	Age of Child	Single Parent	Access on Health System
COVID-19 is caused by a virus.	85%	14.3%	0.000 I **: Other	0.000 I: primary school	0.020 I: retired	0.000 I: None	0.005 I: low	NS *	NS	0.021 I: low
COVID-19 in children causes cold or flu symptoms.	72.3%	26.5%	0.010 I: Other	0.000 I: High school	0.000 I: unemployed	0.006 I: None	0.000 I: low	0,005 I:6–11 years	0.009 I: Yes	NS
Antibiotics are always needed for my child’s COVID-19.	31.6%	67.3%	0.006 I: Other	0.027 I: primary school	0.000 I: student	0.007 I: None	0.000 I: low	0.006 I:12–16 years	NS	NS
My child will be sick for a longer time if it doesn’t get treated with antibiotics for its cold or flu symptoms.	36.9%	62.1%	0.013 I: Other	0.000 I: primary school	0.013 I: retired	0.007 I: None	0.002 I: low	NS	NS	
I am more satisfied if my pediatrician assures me that my child doesn’t need antibiotics for its cold or flu symptoms.	13.8%	85.7%	NS	NS	NS	NS	0.038 I: low	NS	NS	0.000 I: low
How often did your pediatrician recommend antibiotics for your child’s cold or flu symptoms by phone during COVID-19 pandemic?	93.2%	6.3%	0.000 I: Other	0.008 I: Primary school	NS	0.001 I: None	NS	NS	0.001 I: Yes	0.019 I: low

* NS: not statistically significant. ** For statistically significant associations for each statement, the subgroup of parents who answered incorrectly (I) in a higher percentage are noticed in parentheses.

**Table 3 antibiotics-10-00802-t003:** Factor analysis of parental characteristics associated with antibiotic misuse *.

**Group 1 (Eigenvalue 3.6/24% of variation)**	
Knowledge of no use for viral infections	−0.656
Education	−0.642
Income	−0.507
Feeling of adequate information	0.491
Second opinion	0.488
**Group 2 (eigenvalue 1.52/10.2% of variation)**	
Father	0.577
Second opinion	0.431
Knows if child needs antibiotics	0.430
Education	0.365
Income	0.347
**Group 3 (eigenvalue 1.31/8.8% of variation)**	
Father	0.623
Knows when child needs antibiotics	−0.571
Education	0.344
Second opinion	0.370
Feeling of adequate information	0.303
**Group 4 (eigenvalue 1.26/8.4% of variation)**	
Knowledge of no use for viral infections	0.500
Feeling of adequate information	0.419
Age	0.403
Income	−0.360
**Group 5 (eigenvalue 1.1/7.3% of variation)**	
Age	0.617
Knows when the child needs antibiotics	−0.396
**Group 6 (eigenvalue 1.05/7% of variation)**	
Knowledge of no use for viral infections	0.415
Second opinion	−0.415
Knows when the child needs antibiotics	0.400
Age	0.317

* Exploratory dichotomous factor analysis (promax oblique rotation and least-squares extraction method) of parental characteristics associated with antibiotic misuse. Only parameters with a factor loading >0.300 are shown.

## Data Availability

The data presented in this study are available on request from the corresponding author.

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
