# Peer review of "Parental Knowledge, Attitude, and Practices on Antibiotic Use for Childhood Upper Respiratory Tract Infections during COVID-19 Pandemic in Greece"

_antibiotics, 2021, doi:10.3390/antibiotics10070802_

Round 1

Reviewer 1 Report

The study of Oikonomou et al analyzes how the economic and mainly sociocultural parents´ characteristics can determined the antibiotics misuse in children with upper respiratory tract infections. The work is well written and presented, although it is likely to have more relevance in the field of sociology than in that of microbiology.

The main limitation of the study, as it is already indicated by the authors, is the low percentage of parents with low cultural and economic level included. More than half of them are urban with university education.

Specific comments:

Tables 2 and 3 should be combined by removing the least relevant columns. Data omitted in the table could be commented on the text.

In conclusions section should be indicated that pandemic does not seem to influence in the attitude of parents towards the use of antibiotics by their children.  As indicated in the discussion, the results are similar to those obtained both in Greece and in other countries before the appearance of COVID (author references 14, 16, 33, 43)

References should be reviewed and minor errors corrected (e.g. ref 9)

Author Response

The study of Oikonomou et al analyzes how the economic and mainly sociocultural parents´ characteristics can determined the antibiotics misuse in children with upper respiratory tract infections. The work is well written and presented, although it is likely to have more relevance in the field of sociology than in that of microbiology. The main limitation of the study, as it is already indicated by the authors, is the low percentage of parents with low cultural and economic level included. More than half of them are urban with university education.

Thank you for your constructive feedback on our manuscript

Specific comments:     

Tables 2 and 3 should be combined by removing the least relevant columns. Data omitted in the table could be commented on the text.

We have combined tables 2 and 3 to one Table (named Table 2 in the revised version of the manuscript) by omitting the least relevant columns. In the results section we have added the following: Of note, KAP on antibiotic use during COVID-19 was not associated with the parental age and gender, chronic disease in the child and number of children in the family.

In conclusions section should be indicated that pandemic does not seem to influence in the attitude of parents towards the use of antibiotics by their children.  As indicated in the discussion, the results are similar to those obtained both in Greece and in other countries before the appearance of COVID (author references 14, 16, 33, 43)

We would like to thank the reviewer for this useful point. We have added the following sentence in the revised version of the manuscript:  Of note, the pandemic does not seem to have influenced parental views on antibiotic use for their offspring.

References should be reviewed and minor errors corrected (e.g. ref 9)

We have corrected Reference 9 and reviewed all references.

Reviewer 2 Report

This is an interesting manuscript with a large number of participants. However, English language and scientific style should be improved throughout the manuscript. Moreover, there are also few things to improve:

Line 43 - use the full name if you introduce abbreviation for the first time

Line 59 - in line 57 you wrote references as 25-27, why are these 28,29, 30

Line 70 - Table 1 is too big, and it’s most important results are not described in text. Also, N/A should be described at the end of the table. It is usual to use . and not , for numbers (for instance as in line 68)

Line 120 - missing . before new sentence

Line 345 - mistake in the reference

Author Response

This is an interesting manuscript with a large number of participants. However, English language and scientific style should be improved throughout the manuscript.

We would like to thank the reviewer for the constructive criticism on our manuscript. The manuscript has gone an extensive language edit by a native speaker

Moreover, there are also few things to improve:

Line 43 - use the full name if you introduce abbreviation for the first time

We have added the definition for the WHO when the abbreviation is introduced for the first time.

Line 59 - in line 57 you wrote references as 25-27, why are these 28,29, 30

The reference 28, 29, 30 describe the situation in Greece

Line 70 - Table 1 is too big, and it’s most important results are not described in text. Also, N/A should be described at the end of the table. It is usual to use . and not , for numbers (for instance as in line 68)

Following your advice we have shortened the table and we have now used . and not , for decimal points. The most important results of Table 1 are now described in the text of the revised manuscript as following: Of note, the majority of our study participants were women (77.2%), aged between 31-50 years old (88.1%), of Greek nationality (85.7%) and with good self-perceived access to healthcare system(65.8%).

Line 120 - missing . before new sentence

We have added the missing dot

Line 345 - mistake in the reference

We have corrected Reference 9

Reviewer 3 Report

I read with great interest the paper. It is well wrote and with interest focus

Below my suggestions

  1. Introduction: Update data on AMR wordwilde and in your country.
  2. Methods and results are clear
  3. Discussion: discuss better the role of education on this items also for young doctor that can be the future of antibiotic prescriptior (see Italian young doctors' knowledge, attitudes and practices on antibiotic use and resistance: A national cross-sectional survey. J Glob Antimicrob Resist. 2020 Dec;23:167-173.) and discuss also some health policy in other country as possible solution on this issue
  4. Furthermore, add some limitations for your paper
  5. Discuss also the role of social determinant of health in knowledge and the role of social media in media compaing on this issue
  6. Conclusion: give some public health action to fight AMR and correct use of antibiotic in your country

Author Response

I read with great interest the paper. It is well written and with interest focus. Below my suggestions:

  1. Introduction: Update data on AMR worldwide and in your country.

Thank you for this comment.

We have now updated the reference 11 on the latest WHO data and also added the following statement in the revised version of the manuscript: As a result, over the past decade the emergence of antibiotic resistance has been recognized by the World Health Organization (WHO) as one of the ten top global public health threats facing humanity [11].

As for the relevant data in Greece the most recent reports are available by the ECDC and can be found in the references 28, 29 and 30. These ECDC data have been updated in 2019 and there no more  national data available in the field since then.

  1. Methods and results are clear.

Τhank  you for this comment

  1. Discussion: discuss better the role of education on this items also for young doctor that can be the future of antibiotic prescriptior (see Italian young doctors' knowledge, attitudes and practices on antibiotic use and resistance: A national cross-sectional survey. J Glob Antimicrob Resist. 2020 Dec;23:167-173.) and discuss also some health policy in other country as possible solution on this issue

Following your advice we have added the following in the discussion section of the revised version of the manuscript: Educational activities for prescribers should start early on in their career as research so far has highlighted gaps in medical training in the field (45, 46). Novel and effective training methods on all aspects of AMS and AMR should be incorporated in the Medical Curricula worldwide. In addition to education, better prescribing requires expert personnel, practice guideline drafting, and implementation aids, as well as setting clear goals and quantitative targets for quality in prescribing (47). All the above have been used with apparent success in some Northern European countries and should guide relevant interventions in Greece.(47)

We have also added the following extra references

  • Di Gennaro F, Marotta C, Amicone M, Bavaro DF, Bernaudo F, Frisicale EM, Kurotschka PK, Mazzari A, Veronese N, Murri R, Fantoni M. Italian young doctors' knowledge, attitudes and practices on antibiotic use and resistance: A national cross-sectional survey. J Glob Antimicrob Resist. 2020 Dec;23:167-173. doi: 10.1016/j.jgar.2020.08.022. Epub 2020 Sep 21. PMID: 32971291.
  • Efthymiou P, Gkentzi D, Dimitriou G. Knowledge, Attitudes and Perceptions of Medical Students on Antimicrobial Stewardship. Antibiotics (Basel). 2020 Nov 17;9(11):821. doi: 10.3390/antibiotics9110821. PMID: 33213047; PMCID: PMC7698472.
  • Kern WV. Organization of antibiotic stewardship in Europe: the way to go. Wien Med Wochenschr. 2021 Feb;171(Suppl 1):4-8. doi: 10.1007/s10354-020-00796-5. Epub 2021 Feb 9. PMID: 33560499; PMCID: PMC7872948.

  1. Furthermore, add some limitations for your paper

We had made clearer the study limitations in the revised version of the manuscript : 

With regards to study limitations, firstly the distribution of the questionnaires took place in summer months of 2020 (during precautionary measures to prevent spread of COVID-19) with low number of primary care visits [32]. Secondly, it was difficult for parents with low socioeconomic status or belonging to special population groups (e.g. Roma), to take part in the study because they might have been uncapable of reading or understanding the questionnaire. Moreover, the study enrolled parents who only sought care at health care centers which limits the generalizability of the results. Finally, data were collected from a large city in Greece and thus mainly represent the urban but not necessarily the rural population.

  1. Discuss also the role of social determinant of health in knowledge and the role of social media in media campaign on this issue

Following your comment we have inserted the following paragraph in the discussion section of our revised manuscript: Our study population included a small percentage of parents with low cultural and economic level. In particular, more than half of our study participants were urban with university education. As the problem of antimicrobial misuse has been also characterized as a social issue it therefore demands a relevant approach from that perspective. Hence, good level of knowledge on antibiotics use could be partially related to the high socioeconomic status of our participants. The involvement of social scientists in the field is crucial to address the multiple dimensions of the problem such as socio-cultural, economic and political. In addition to the above, given the dynamic and constantly changing nature of human behavior social media could also play an important role in public health interventions to combat antimicrobial resistance and their role is yet to be further established. 

We have also added the following extra references:

  • Lu J, Sheldenkar A, Lwin MO. A decade of antimicrobial resistance research in social science fields: a scientometric review. Antimicrob Resist Infect Control. 2020 Nov 4;9(1):178. doi: 10.1186/s13756-020-00834-2. PMID: 33148344; PMCID: PMC7643349.
  • Acharya KP, Subedi D. Use of Social Media as a Tool to Reduce Antibiotic Usage: A Neglected Approach to Combat Antimicrobial Resistance in Low and Middle Income Countries. Front Public Health. 2020 Dec 10;8:558576. doi: 10.3389/fpubh.2020.558576. PMID: 33363074; PMCID: PMC7758238.
  • Davis M, Whittaker A, Lindgren M, Djerf-Pierre M, Manderson L, Flowers P. Understanding media publics and the antimicrobial resistance crisis. Glob Public Health. 2018 Sep;13(9):1158-1168. doi: 10.1080/17441692.2017.1336248. Epub 2017 Jun 8. Erratum in: Glob Public Health. 2018 Sep;13(9):i. PMID: 28594309.

  1. Conclusion: give some public health action to fight AMR and correct use of antibiotic in your country

Following your comment we have inserted the following paragraph in the conclusion section of our revised manuscript.

Furthermore, studies assessing country-specific determinants of antibiotic misuse, such as country wealth and health care system particularities would also be useful for the implementation of multilevel interventional programs aimed at limiting the spread of antibiotic resistance. Despite multiple efforts to achieve that in Greece during the last decades there is still room for further development. Public health interventions at a national level should be constant and sustainable following the successful examples of other European countries.

Round 2

Reviewer 2 Report

The manuscript has been improved accordingly.